# Validation of conventional PCR-like alternative to SARS-CoV-2 detection with target nucleocapsid protein gene in naso-oropharyngeal samples

**Rogério Fernandes Carvalho**[1☯*], **Monike da Silva Oliveira**[2☯], **Juliane Ribeiro**[3‡], **Isac Gabriel Cunha dos Santos**[1☯], **Katyane de Sousa Almeida**[1‡], **Ana Carolina Muller Conti**[1☯], **Bruna Alexandrino**[1‡], **Fabrício Souza Campos**[4‡], **Célia Maria de Almeida Soares**[2‡], **José Carlos Ribeiro Júnior**[1☯]

**1** Laboratory of Microbiology, Federal University of North Tocantins, Araguaína, Tocantins, Brazil, **2** Laboratory of Molecular Biology, Federal University of Goiás, Goiânia, Goiás, Brazil, **3** Laboratory of Virology and Molecular Biology, State University of Londrina, Londrina, Paraná, Brazil, **4** Laboratory of Bioinformatics and Biotechnology, Federal University of Tocantins, Gurupi, Tocantins, Brazil

☯ These authors contributed equally to this work.
‡ These authors also contributed equally to this work
* fcrogeriofc@hotmail.com

**Data Availability Statement:** All relevant data are within the manuscript and its Supporting information files.

## Abstract

SARS-CoV-2 has spread worldwide and has become a global health problem. As a result, the demand for inputs for diagnostic tests rose dramatically, as did the cost. Countries with inadequate infrastructure experience difficulties in expanding their qPCR testing capacity. Therefore, the development of sensitive and specific alternative methods is essential. This study aimed to develop, standardize, optimize, and validate conventional RT-PCR targeting the N gene of SARS-CoV-2 in naso-oropharyngeal swab samples compared to qPCR. Using bioinformatics tools, specific primers were determined, with a product expected to be 519 bp. The reaction conditions were optimized using a commercial positive control, and the detection limit was determined to be 100 fragments. To validate conventional RT-PCR, we determined a representative sampling of 346 samples from patients with suspected infection whose diagnosis was made in parallel with qPCR. A sensitivity of 92.1% and specificity of 100% were verified, with an accuracy of 95.66% and correlation coefficient of 0.913. Under current Brazilian conditions, this method generates approximately 60% savings compared to qPCR costs. Conventional RT-PCR, validated herein, showed sufficient results for the detection of SARS-CoV-2 and can be used as an alternative for epidemiological studies and interspecies correlations.

## Introduction

Coronavirus (CoV) belongs to the family *Coronaviridae* and is found in various animals and humans. In late 2019 and early 2020, the etiological agent of the new Wuhan pneumonia in

**Funding:** This work was supported by the Pro-Reitoria de Pesquisa of the Federal University of Tocantins, Prefeitura Municipal de Araguaína, Tocantins, Brazil and by Postgraduate Program in Animal Health and Public Health of Federal University of North Tocantins. JCRJ was the only recipient of this financial support. This study was also partially funded by the researchers.

**Competing interests:** The authors have declared that no competing interests exist.

China was identified as severe acute respiratory syndrome coronavirus 2 (SARS-CoV-2) [1]. Since then, SARS-CoV-2 has spread worldwide and has become a global health problem, causing coronavirus disease 2019 (COVID-19). As of August 7, 2021, more than 562,000 people have died in Brazil due to complications of COVID-19, being the country with the second highest number of deaths. Worldwide, the 20 countries with the highest death counts have collectively more than 3 million deaths [2].

SARS-CoV-2 is a simple positive-sense enveloped RNA virus that encodes the spike (S), envelope (E), membrane (M), and nucleocapsid (N) as structural proteins [1]. Coronavirus nucleocapsid protein has a structural role as it forms a ribonucleoprotein complex with gRNA and also has RNA chaperone activity [3]. This protein also has an important antigenic action in the formation of specific antibodies in natural infection [4].

The World Health Organization (WHO) recommends the use of real-time PCR (qPCR) for the diagnosis of SARS-CoV-2, using different protocols aimed at detecting genes of non-structural and structural proteins, especially the N gene [5].

However, qPCR inputs and equipment are notoriously scarce with the increase in worldwide demand, especially in countries dependent on imports of these resources [6, 7], such as Brazil. This difficulty in accessing diagnostic resources leads to the underreporting of cases [8]. Countries that implemented greater diagnostic coverage in the monitoring of their populations were able to control the dispersion of the disease more efficiently [9].

Some alternatives to qPCR have been proposed by different researchers and may present sufficient performance to increase the supply of diagnostic resources for SARS-CoV-2 in countries with inadequate infrastructure [7], or for studies of interspecies epidemiological correlations that are yet unclear. Conventional PCR preceded by reverse transcription may be one such alternative [7]. Therefore, the present study aimed to develop, standardize, optimize, and validate a conventional RT-PCR method targeting the N gene of SARS-CoV-2 in nasopharyngeal and oropharyngeal swab samples.

And in the end, the conventional RT-PCR for the diagnosis of SARS-CoV-2 proved to be an alternative tool for expanding population testing, with significant sensitivity and specificity.

## Material and methods

Primers were established using the genomic sequence available in GenBank (accession number MT081066) for the N gene, according to the protocols developed and recommended by the WHO for detection of SARS-CoV-2, using Primer 3 (http://primer3.ut.ee/) and Primer Blast (https://www.ncbi.nlm.nih.gov/tools/primer-blast/) programs. Two sets of forward and reverse primers were evaluated, the most sensitive being N1-F (5′–GGTTCACCTCTCTCACTC AA–3′) and N2-R (5′–CAAGCAGCAGCAAAAGCAAGA–3′), with an expected product of 519 bp.

Amplification conditions were optimized in a gradient (T100 Thermal Cycler, Bio-Rad, Hercules, CA, USA) with a positive commercial control for the N gene (2019-NCOV_N Positive Control Kit, Integrated DNA Technologies, Coralville, IA, USA; 200,000 copies/μL) and determined with cycle conditions of 94˚C for 5 min; 40 cycles of 94˚C for 45 s, 52 ºC for 45 s, and 72 ºC for 1 min; and a final extension cycle at 72 ºC for 10 min. To determine the detection limit of the proposed method, commercial positive control aliquots were serially diluted in TE buffer (Tris-HCl [10mM]: EDTA [1 mM]). PCR products were subjected to electrophoresis in 2% agarose, followed by ethidium bromide staining (0.2 mg/mL). The readings were performed under ultraviolet light.

In the validation trial, the sample size was determined for 346 volunteers with suspected COVID-19 infection. The calculation of the sample universe was performed using EpiInfo

7.2.4 (CDC, Atlanta, GA, USA) with a 95% confidence interval, 5% standard error, 90% estimated sensitivity, and 10% disease prevalence among suspected cases (epidemiological data made available by the municipality of Araguaína, Tocantins, Brazil, on May 4, 2020), and the population of the municipality of Araguaína was estimated at 180,470 inhabitants [10].

Samples were collected from August 24 to September 10, 2020. The participants of this study were classified as suspected to have COVID-19 by the municipal medical service and were referred for sample collection for official diagnosis in Tocantins. Three rayon swabs (bilateral nasopharyngeal and simple oropharyngeal) were collected, stored in a Falcon tube with 3 mL of saline solution at 0.9%, and refrigerated at 2–8˚C.

The official diagnosis involved qPCR (gold standard) performed at the COVID-19 Diagnostic Support Unit of the Oswaldo Cruz Foundation (FIOCRUZ) in Rio de Janeiro, Brazil, using the Berlin Protocol. Patients with Ct $\leq$ 40 for the E gene and Ct $\leq$ 35 for the RdRp region were considered COVID-19-positive.

Patients referred for official diagnosis were invited to voluntarily participate in this research. For this, a second set of samples was collected from the same patients using the same collection procedure. The samples were sent to the Microbiology Laboratory of the Federal University of North Tocantins, Araguaína University campus, where they were processed within 24 h.

RNA extraction was performed using a commercial kit (QIAamp Viral RNA Mini Kit, Qiagen, Valencia, CA, USA) in an NB2 biological safety booth. The extracted product was reverse transcribed using a commercial kit (SuperScript III First-Strand Synthesis System, Invitrogen, Carlsbad, CA, USA) according to the manufacturer's protocol. The RT products (4 μL) were subjected to PCR with a commercial kit (Platinum Hot-Start PCR Master Mix, Invitrogen) together with primers (20 pmol) established for the N gene of SARS-CoV-2. The reactions were amplified according to the optimized protocol with the commercial positive control and individually paired with a positive control reaction at a concentration of 1,000 fragments.

For statistical correlation of agreement between the methods (qPCR and conventional RT-PCR), the Kappa test was performed using BioEstat 5.3. Samples positive for RT-PCR were randomly selected for sequencing by the Sanger method (ABI 3500 Genetic Analyzer, Applied Biosystems, Foster City, CA, USA) in both directions after purification (Wizard SV Gel and PCR Clean-Up System, Promega, Southampton, UK) of the amplified products. Sequence quality was evaluated using BioEdit v. 7.2.5 [11], and consensus sequences were generated by CAP 3 [12]. These sequences were individually aligned using Clustal W and the representative sequences from GenBank, and genetic similarity was analyzed via the neighbor-joining method and the Tamura-Nei model using 1,000 bootstrap replications in MEGA X software [13].

The study was submitted and approved by the Brazilian National Committee of Ethics and Research (no. 33350720.9.0000.5519) before the research was conducted. Only older patients ($\geq$18 years of age) were included in the study and the consent was verbatim attested to when signing the Informed Consent Form according with Brazilian National Committee of Ethics.

## Results and discussion

In determining the detection limit, it was verified that RT-PCR was sensitive to 100 copies of the N gene. Therefore, the proposed method has good sensitivity even though it has a higher detection limit than qPCR. This detection capability is considered satisfactory because the viral load in the naso-oropharyngeal cavities is high and significant. Previous studies have shown 17,429 copies in scars, 2,552 copies in pharyngeal swabs, and 651 copies in nasal swabs

**Table 1. Comparison between the results of samples processed by qPCR and samples processed by conventional RT-PCR.**

| | | Gold Standard (*q*PCR) | | |
| --- | --- | --- | --- | --- |
| | | Positive | Negative | Total |
| **Conventional RT-PCR** | Positive | 179 | 0 | 179 |
| | Negative | 15 | 152 | 167 |
| | Total | 194 | 152 | 346 |

[14]. Another study showed viral load among positive patients ranging from 641 to $1.34 \times 10^{11}$ copies [15].

In qPCR, with a target in the RdRp region, 3.8 copies of RNA can be detected [16]. In LAMP, the limit of detection can reach one copy of viral RNA, while Penn-RAMP studies can detect as few as seven copies of viral RNA per reaction [17]. In studies with multiplex qPCR (N1 and N2), it was possible to detect as little as five copies per reaction [18]. In another study using qRT-PCR, the detection limit for the N gene was 10 copies per reaction [19]. Thus, it was expected that, in patients with a low viral load in nasopharyngeal or oropharyngeal samples or in cases with a problem that compromised the quality and amplitude of the collection, the sensitivity of conventional RT-PCR would be lower than that of qPCR.

Of the 346 clinical samples of patients tested by qPCR, 194 were detectable and 152 were non-detectable. The analysis using conventional RT-PCR was in agreement with the gold standard in 331 samples. Among the 15 discordant samples, 15 false-negative and non-false-positive samples were observed, as shown in Table 1.

The Brazilian legislation for validation of qualitative analytical methods of identification [20] only indicated a criterion for selectivity. The results showed that the specificity of conventional RT-PCR was 100%, the sensitivity was 92.27%, and the accuracy was 95.66%. The false-positive and false-negative rates were 0% and 8.98%, respectively (Table 2). A preliminary study conducted in 2020 in Brazil with 116 samples, which compared conventional RT-PCR with qPCR, showed 92.1% sensitivity and 100% specificity for conventional RT-PCR [7].

Four samples were randomly selected for sequencing the amplified products. All products were confirmed as SARS-CoV-2 (GenBank accession numbers MW281570, MW281571, MW281572, and MW281573). The phylogenetic tree of Brazilian samples, in relation to those deposited in GenBank in different countries, is shown in Fig 1. Thus, the method presented 100% specificity among the study samples, demonstrating the ability of the method to not detect other possible viruses or contaminants in the matrix evaluated in the present study.

Upon correlating the methods, a coefficient of 0.9129 was obtained. According to Landis and Koch [21], values between 0.81 and 1.00 indicate a high correlation. This coefficient indicated 95.66% agreement between the methods. Thus, conventional RT-PCR for the diagnosis of SARS-CoV-2, developed and validated by the present study, can be used as an alternative

**Table 2. Calculations of the validation parameters of classical RT-PCR in relation to the qPCR for SARS-CoV-2.**

| | Evaluation of Conventional RT-PCR | |
| --- | --- | --- |
| **False-positive rate** | No | (0/152) |
| **False-negative rate** | 8.98% | (15/167) |
| **Specificity** | 100% (CI: 97.54%~100%) | (152/152) |
| **Sensitivity** | 92.27% (CI: 87.64%~95.26%) | (179/194) |
| **Accuracy** | 95.66% | (179+152/346) |
| **Detection threshold** | 100 copies | - |

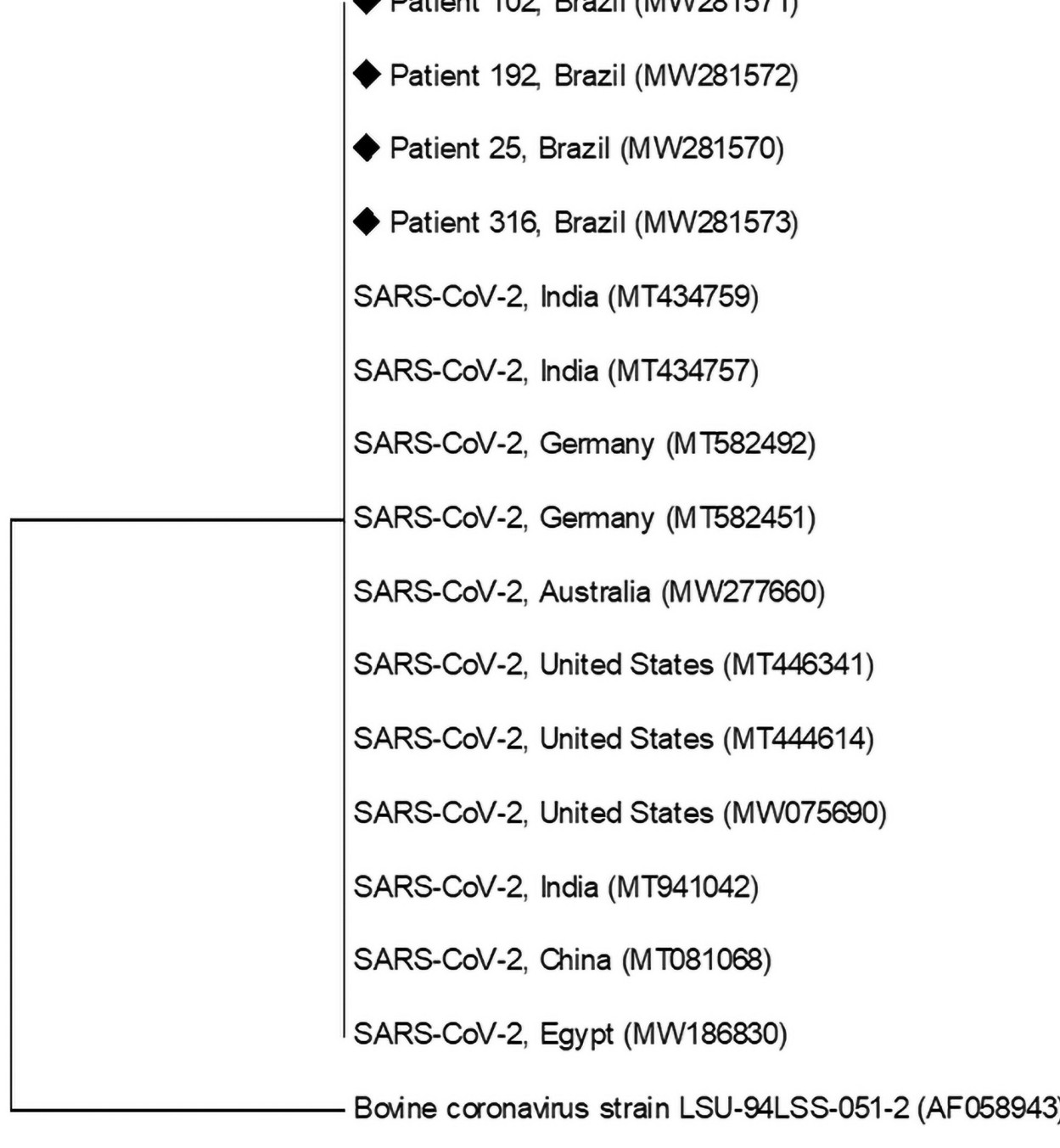

**Fig 1. Phylogenetic tree of N gene sequences from SARS-CoV-2 isolated in several countries and Brazil (◆).**

tool for expanding population testing, especially in places lacking the infrastructure for qPCR. According to current costs in Brazil, the test proposed herein costs 32% of the value of qPCR. This method can also be an alternative tool for future studies on unclear epidemiological correlations between humans and animals, allowing the allocation of qPCR resources for the diagnosis of humans.

## Conclusion

The diagnostic method for identifying COVID-19 through conventional RT-PCR was developed, optimized, and validated in terms of specificity and sensitivity using the N gene, and it produced an adequate yield for use in qualitative diagnosis as an alternative to qPCR. As the study used simple and easily acquired commercial kits, easy reproducibility and considerable cost savings are expected.

## Supporting information

**S1 Dataset.**
(XLS)

## Author Contributions

**Conceptualization:** Rogério Fernandes Carvalho, Monike da Silva Oliveira, Juliane Ribeiro, Katyane de Sousa Almeida, Ana Carolina Muller Conti, Bruna Alexandrino, Fabrício Souza Campos, Célia Maria de Almeida Soares, José Carlos Ribeiro Júnior.

**Data curation:** Monike da Silva Oliveira, Fabrício Souza Campos, José Carlos Ribeiro Júnior.

**Formal analysis:** Rogério Fernandes Carvalho, Monike da Silva Oliveira, Isac Gabriel Cunha dos Santos, Célia Maria de Almeida Soares, José Carlos Ribeiro Júnior.

**Funding acquisition:** Monike da Silva Oliveira, José Carlos Ribeiro Júnior.

**Investigation:** Rogério Fernandes Carvalho, Monike da Silva Oliveira, José Carlos Ribeiro Júnior.

**Methodology:** Rogério Fernandes Carvalho, Monike da Silva Oliveira, Juliane Ribeiro, Isac Gabriel Cunha dos Santos, Katyane de Sousa Almeida, Ana Carolina Muller Conti, Bruna Alexandrino, José Carlos Ribeiro Júnior.

**Project administration:** Rogério Fernandes Carvalho, Monike da Silva Oliveira, Isac Gabriel Cunha dos Santos, José Carlos Ribeiro Júnior.

**Resources:** Monike da Silva Oliveira, José Carlos Ribeiro Júnior.

**Software:** Rogério Fernandes Carvalho, Monike da Silva Oliveira, José Carlos Ribeiro Júnior.

**Supervision:** Rogério Fernandes Carvalho, Monike da Silva Oliveira, Juliane Ribeiro, Katyane de Sousa Almeida, Ana Carolina Muller Conti, Bruna Alexandrino, José Carlos Ribeiro Júnior.

**Validation:** Rogério Fernandes Carvalho, Monike da Silva Oliveira, José Carlos Ribeiro Júnior.

**Visualization:** Rogério Fernandes Carvalho, Monike da Silva Oliveira, Juliane Ribeiro, Katyane de Sousa Almeida, Ana Carolina Muller Conti, Bruna Alexandrino, Célia Maria de Almeida Soares, José Carlos Ribeiro Júnior.

**Writing – original draft:** Rogério Fernandes Carvalho, Monike da Silva Oliveira, José Carlos Ribeiro Júnior.

**Writing – review & editing:** Rogério Fernandes Carvalho, Monike da Silva Oliveira, Juliane Ribeiro, Katyane de Sousa Almeida, Ana Carolina Muller Conti, Bruna Alexandrino, Fabrício Souza Campos, Célia Maria de Almeida Soares, José Carlos Ribeiro Júnior.

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
