## [Decision Letter · Decision Letter 0]

19 Aug 2021

PONE-D-21-11819

Validation of conventional PCR-like alternative to SARS-CoV-2 detection with target nucleocapsid protein gene in naso-oropharyngeal samples

PLOS ONE

Dear Dr. Carvalho,

Thank you for submitting your manuscript to PLOS ONE. After careful consideration, we feel that it has merit but does not fully meet PLOS ONE’s publication criteria as it currently stands. Therefore, we invite you to submit a revised version of the manuscript that addresses the points raised during the review process.

Specifically, please summarize the NC functions and the immune responses in natural infections. Relevant references on the viral nucleocapsid N protein are from Luis Enjuanes et al. showing that N is a multifunctional protein with RNA binding and chaperoning activities. In addition specific IGG against N are found in people who experienced Cov infection.

We look forward to receiving your revised manuscript.

Kind regards,

Jean-Luc EPH Darlix, MG, Ph.D.

Academic Editor

PLOS ONE

Journal Requirements:

Additional Editor Comments (if provided):

Reviewers' comments:

Reviewer's Responses to Questions

**Comments to the Author**

1. Is the manuscript technically sound, and do the data support the conclusions?

Reviewer #1: Yes

2. Has the statistical analysis been performed appropriately and rigorously? 

Reviewer #1: Yes

3. Have the authors made all data underlying the findings in their manuscript fully available?

Reviewer #1: Yes

4. Is the manuscript presented in an intelligible fashion and written in standard English?

Reviewer #1: Yes

5. Review Comments to the Author

Reviewer #1: This is an interesting paper that describes an alternative, PCR-based method for COVID-19 diagnosis, beyond the predominant real-time PCR methods that are being utilized worldwide. While the total amount of data presented and overall scope of the work is limited, the main point of the paper is important and timely. Thus, while this manuscript is much shorter than most full-length articles, I recommend publishing, due to the importance of the findings.

6. PLOS authors have the option to publish the peer review history of their article (what does this mean?). If published, this will include your full peer review and any attached files.

Reviewer #1: No

---

## [Author Response · Author response to Decision Letter 0]

24 Aug 2021

Editor Comments:

Dear Dr. Jean-Luc Darlix, 

Thank you for your availability and attention in editing our manuscript.

We fully accept the suggestions and respond to all, from the editor and reviewer.

We also include as an author, in this review version, Prof. Fabrício Campos, PhD, a virologist who also contributed to the study.

We do not find it necessary to deposit our protocol in the recommended domain (protocols.io) since our methodology is very descriptive and allows for full reproducibility of all analyzes performed.

A peer-reviewed Lab Protocol article submission we also think it would frame as self-plagiarism.

To avoid any problems in writing in English, we forward the file to a company specialized in academic review. The revision certificate is attached.

We update epidemiological data from Brazil in the introduction.

Specific comments are identified by the authors (AU):

Specifically, please summarize the NC functions and the immune responses in natural infections. Relevant references on the viral nucleocapsid N protein are from Luis Enjuanes et al. showing that N is a multifunctional protein with RNA binding and chaperoning activities. In addition specific IGG against N are found in people who experienced Cov infection.

AU: We have fully included your suggested and reference in the introduction, Line 55 to 55, highlighted in yellow.

Reviewer 1: 

This is an interesting paper that describes an alternative, PCR-based method for COVID-19 diagnosis, beyond the predominant real-time PCR methods that are being utilized worldwide. While the total amount of data presented and overall scope of the work is limited, the main point of the paper is important and timely. Thus, while this manuscript is much shorter than most full-length articles, I recommend publishing, due to the importance of the findings.

AU: Dear reviewer, thank you for your evaluation.

We are happy for the recognition of the merit of our laborious study, executed under many difficulties in north of Brazil in the current situation.

Our goal has always been the most objective and accurate wording possible. This justifies our shorter format, making it easier to read and interpret the result for the reader.

All authors are grateful for your valuable and important collaboration in evaluation of the manuscript.

---

## [Editor Report · Decision Letter 1]

31 Aug 2021

Validation of conventional PCR-like alternative to SARS-CoV-2 detection with target nucleocapsid protein gene in naso-oropharyngeal samples

PONE-D-21-11819R1

Dear Dr. Carvalho,

We’re pleased to inform you that your manuscript has been judged scientifically suitable for publication and will be formally accepted for publication once it meets all outstanding technical requirements.

Kind regards,

Jean-Luc EPH Darlix, MG, Ph.D.

Academic Editor

PLOS ONE
---

## [Editor Report · Acceptance letter]

16 Sep 2021

PONE-D-21-11819R1 

Validation of conventional PCR-like alternative to SARS-CoV-2 detection
with target nucleocapsid protein gene in naso-oropharyngeal samples 

Dear Dr. Carvalho:

I'm pleased to inform you that your manuscript has been deemed suitable for publication in PLOS ONE. Congratulations! Your manuscript is now with our production department. 

Kind regards, 

on behalf of

Professor Jean-Luc EPH Darlix 

Academic Editor

PLOS ONE